# Multicomponent Low Initial Molar Ratio of SiO_2_/Al_2_O_3_ Geopolymer Mortars: Pilot Research

**DOI:** 10.3390/ma15175943

**Published:** 2022-08-28

**Authors:** Barbara Słomka-Słupik, Paulina Wiśniewska, Wiktor Bargieł

**Affiliations:** 1Department of Building Structures, Faculty of Civil Engineering, Silesian University of Technology, 44-100 Gliwice, Poland; 2Faculty of Civil Engineering, Silesian University of Technology, 44-100 Gliwice, Poland

**Keywords:** geopolymer, alkali activation, anthropogenic raw materials, PV glass powder, kaolin clay, alumina-lime cement, autoclaved cellular concrete, slag, fly ash

## Abstract

Alkali-activated binders have the potential to consume various types of waste materials. Low initial molar ratios of SiO_2_/Al_2_O_3_ geopolymer mortars were considered in this article. Here we studied alkali-activated binders produced with photovoltaic glass powder in 5%; kaolin clay in 15%; ground granulated blast furnace slag in 30%; alumina-lime cement in 30%; and, interchangeably, fly ash from coal combustion in 5%, fly ash from biomass combustion in 5%, or granulated autoclaved cellular concrete in 5%. The influence of clay dehydroxylation, curing conditions, glass presence, and a kind of waste material was investigated. According to the experimental results, strength (compressive and tensile) gradually increased with increasing time and with the use of calcined clay. Significant improvement in compressive strength was seen with the additional 3 days curing time in 105 °C when non-sintered clay was used. The presence of photovoltaic glass in alkali-activated mortars immobilised mercury and arsenic but released zinc, chromium, and sulphates. The microscopic observations confirmed the greater densification of the microstructure of the binder made of calcined clay due to its greater surface development and dehydroxylation. The binder of non-calcined clay was granular, and the interfacial transitional zone was more porous. The C–A–S–H gel seemed to be the main phase. XRD examination confirmed the presence of C–A–S–H, C–S–H, zeolites, and many other phases in minor amount. The presented research was a pilot study, and its main goal was to develop it further.

## 1. Introduction

In this article, the use of different secondary materials in the mass of a binder or building mortar to perform a multicomponent low initial molar ratio of SiO_2_/Al_2_O_3_ geopolymers was considered. Particular attention was paid to waste glass from photovoltaic (PV) panels, as few studies indicated its use as a geopolymer mass component. According to the IRENA report from 2016 [1], it was expected that, by 2050, the amount of photovoltaic modules that reach the end of their life will increase to 5.5–6 million tons. However, now, according to the brochure [2] shown in Figure 1, 1.7 to 8 million tonnes of cumulative PV waste will be generated by the end of 2030 and 60 to 80 million tonnes by the end of 2050. The last report [3] from November 2019 noted the increased awareness and research on end-of-life management of solar PV in the field of reducing, reusing, and recycling.

Recently, due to the trends in the world in the field of decarbonisation [4,5,6,7,8,9,10], many scientists are conducting research on cementless binders [11,12,13,14,15,16]. Most often, these are geopolymer binders [17,18,19,20,21,22,23,24,25,26,27,28,29,30]. Some of them use glass as a component: waste cathode ray tube (CRT) glass [20,21], liquid-crystal display (LCD) glass [22], and glass powder (GP) [23,24,25,26,27]; as well as huge amounts of municipal recycled glass, industrial recycled glass, glass derived from lighting equipment, borosilicate glass from pharmaceutical package, fluorescent lamps, glass produced by DC plasma treatment of waste, and even thin-film transistor liquid display panels [28]. However, there are not many scientific articles that exactly indicate the use of glass from photovoltaic panels as a component of binders. Skripkiūnas et al. [31] investigated the properties of concrete containing various quantities of copper indium selenide (CIS) solar module waste by replacing a certain part (up to 40%) of sand and observed that the compressive strength of the specimens was higher when the sand aggregate was replaced by CIS solar module waste particles from 5 to 20%. Máčalová et al. [32] replaced 100% of natural aggregates with recycled glass from PV panels in concrete made of Portland cement. The awareness of the harmful effect of glass on cement concrete, which is related to the alkali–silica reaction (ASR), is commonly present [17,33,34]. Hao et al. [35] examined the compressive strengths of metakaolinite-based geopolymer with solar panel waste glass after 28 days of curing. The geopolymer samples contained solar panel waste glass in the amount of 10 and 20% of the total mass. The strength was higher, 63 MPa, when 10% of glass was used and 49 MPa when 20% of glass was used. Several studies showed that waste glass in powder form can be successfully incorporated as supplementary cementitious material both in alkaline-activated and cement binders. Using different types of waste glass in geopolymer production is caused by the amorphous nature and pozzolanic properties of these materials [27,28].

Geopolymerisation is mainly a developing field of research for utilising solid waste and by-products. From a chemical point of view, it is a geosynthesis of naturally occurring silico-aluminates or other pozzolanic compounds, silica, and alumina source that can dissolve in the alkaline solution—it can act as a source of geopolymer precursors and thus is able to undergo geopolymerisation [36]. The alkaline solution or component is an activator; it is a compound of the elements of the first group in the periodic table, mainly NaOH and KOH. Products of geopolymerisation are also used as immobilisers of metals [37,38,39,40,41,42]. Setting of the geopolymeric mixture quickly occurs without time for the formation of a proper crystal structure resulting in a microcrystalline, amorphous, or semi-amorphous structure depending on the reaction conditions [43]. Undoubtedly Davidovits is considered the precursor of this field of science even before the 1980s [44]. Davitovits stated that the satisfactory properties of the synthesised products can be obtained if the following proportions are kept: M_2_O/SiO_2_, 0.2–0.48; SiO_2_/Al_2_O_3_, 3.3–4.5; H_2_O/M_2_O, 10–25; and M_2_O/Al_2_O_3_, 0.8–1.6 [36,45]. However, according to Van Jaarsveld [43], based on [46], Davidovits and co-workers indicated that certain compositional criteria have to be met for geopolymerisation to occur. These include the following: the molar ratio SiO_2_:M_2_O must be between 4.0:1 and 6.6:1 in the aqueous soluble silicate solution, where M is an alkali metal cation; the aluminosilicate oxide must contain Al, which is readily soluble; and the overall molar ratio Al_2_O_3_:SiO_2_ must be between 1:5.5 and 1:6.5. As can be seen, these criteria are not unambiguous; therefore, after [43], it must be assumed that while using waste reactants for the reaction, one should develop his or her own criteria that are adequate to these substrates.

The geopolymeric aluminosilicate has been grouped in three families depending on the atomic ratio Si/Al, which may be 1, 2, 3, or >3 [36,45]. Amorphous to semi-crystalline three-dimensional aluminosilicate structures have been listed: the poly(sialate) type (Si–O–Al–O–), poly(sialate-siloxo) type (Si–O–Al–O–Si–O–), and poly(sialate-disiloxo) type (Si–O–Al–O–Si–O–Si–O–) [44]. However, it should be noted that Provis and Van Deventer [18] mentioned that the sialate nomenclature of Davidovits “implies certain aspects of the geopolymer gel structure which do not correspond to reality”.

The general mechanism for the alkali activation of materials primarily comprising silica and reactive alumina was proposed by Glukhovsky in the 1950s [47]. The mechanism of the Glukhovsky model is composed of simultaneous reactions of destruction–coagulation–condensation–crystallisation. The first step consists of a breakdown of the covalent bonds Si–O–Si and Al–O–Si, which happens when the pH of the alkaline solution rises, so those groups are transformed into a colloid phase. Then an accumulation of the destroyed products occurs, which interacts among them to form a coagulated structure, leading in a third phase to the generation of a condensed and crystallised structure.

One of the most important source materials for geopolymer is fly ash [15,19,20,26,28]. It is a secondary material, is waste from energy processes, is generally available, and comes from the combustion of coal and/or biomass [13,28,48,49]. The ash from the combustion of hard coal is mainly used in the production of cement [19,28]. It contains large amounts of reactive silica, but the amount of calcium is too small for the pozzolanic reactions to occur when the ash comes into contact with water [19]. The geopolymerisation process consists of exciting aluminium and silicon atoms with a strongly alkaline compound so that it is dissolved to form a geopolymer paste, of which the main phase is the C–S–H gel [18,30,50].

Another precursor of geopolymers is clay [24,25,51,52,53,54]. Similar to fly ash, clay minerals are diverse and the resources of aluminosilicates are available worldwide [51]. The use of clays in the production of mechanically strong geopolymers requires its preliminary treatment. This can be accomplished by thermal, mechanical, or chemical means [51]. Better pozzolana properties are obtained by burning; loss of combined water and OH-groups causes the crystalline network of the clay mineral to be destroyed, while silica and alumina remain in an unstable amorphous state that reacts with calcium hydroxide [52]. The most popular in the production of geopolymers is metakaolin, which is obtained during calcinations of kaolinitic clays at temperatures ranging from 500 to 800 °C, depending on the clay [52,53,54].

Alkaline activation must also include the ground granulated blast furnace slag (GGBFS). The chemical component of GGBFS mainly consists of the CaO–SiO_2_–MgO–Al_2_O_3_ system, gehlenite (2CaO·Al_2_O_3_·SiO_2_), akermanite (2CaO·MgO·2SiO_2_), and depolymerised calcium aluminosilicate glass. The glass content of the slag should be in excess of 90% to show satisfactory properties. The major binding phase in alkali-activated ground granulated blast furnace slag (AA GGBFS) is the C–S–H gel, unlike geopolymers, which are assumed to be the result of the formation of three-dimensional zeolite such as polymers [55].

In order to increase the amount of lime and aluminium in geopolymers, alumina-lime cement (ALC) can be used. Lime is needed for pozzolanic reactions, and aluminium in the available secondary materials is relatively little, taking into account silicon. It was found that calcium has a positive effect on the comprehensive strength of geopolymeric binds, and the formation of Ca compounds in geopolymers is greatly dependent on the pH and Si/Al ratio [56,57]. The addition of a sufficient quantity of Ca to geopolymers in the form of calcium hydroxide can lead to the formation of phase-separated Al-substitute calcium silicate hydrate (C–(A)–S–H) and geopolymer (N–A–S–H) gels [55]. Li et al. [55] emphasised also that Ca^2+^ is capable of acting as a charge-balancing cation within the geopolymeric binding structure. On the other hand, the role of aluminium is also significant because its availability controls the properties of geopolymers, such as setting characteristics, flexural strength, acid resistance, microstructure, and strength development profile [55].

The curing process for geopolymers is also of great importance. Complete geopolymerisation reactions were observed when curing at 40–85 °C [58]. Most studies confirmed the increase in the strength of geopolymers matured at higher temperature; however, Gołek and Deja [59] noticed that accelerating hydration by increasing the temperature or pressure results in the creation of favourable conditions for the growth of crystalline phases, which is the main reason for the observed drops in strength.

Various components have been and still are used in the production of geopolymers, which are examined in terms of composition, curing temperature, alkali concentration, water/solid ratio, strength, shrinkage property, performance in chemical exposure, resistance to freeze–thawing effect or aggressive attack, and heat resistance. To summarise, precursors can be materials rich in Si (such as fly ash or slag) and materials rich in Al (clays) naturally occurring, can come from industrial processes, and can be categorised as waste. The commonly used activators, in turn, are as follows: NaOH, Na_2_SO_4_, waterglass, Na_2_CO_3_, K_2_CO_3_, KOH, K_2_SO_4_, or a little amount of cement clinker and others [36,60,61,62,63,64].

The aim of the article was to compare the mechanical and material properties of mortars made on standard sand to determine the strength of the binders. These mortars were made of components containing aluminates and silicates, which bind after activation with an alkaline sodium solution. The properties were compared depending on the kaolin clay (KC)—fired and unfired—in the mixture. The influence of the secondary material was also highlighted. However, the amount of aluminium was increased in the mixtures. The value of SiO_2_/Al_2_O_3_ in the designed blends was low, around 2 (Si/Al~1), so poly(sialate) type (–Si–O–Al–O–) geopolymer should be considered.

## 2. Methods and Materials

### 2.1. Methods

#### 2.1.1. XRF: X-ray Fluorescence Method

X-ray fluorescence (XRF) is a common method used in the cement industry to test the chemical composition of cement and other additives and powders. Tests were carried out according to PN-EN ISO 12677:2011 [65] on samples grounded to a grain size less than 100 μm and dried at 105 °C to constant weight. Dried samples were ignited at 1025 °C, and the loss of ignition was determined. In order to prepare the fused cast bead used for analysis, the sample ignited to constant weight was fused with a mixture of lithium tetraborate (66.67%), lithium metaborate (32.83%), and lithium bromide (0.5%) produced by Spex CertiPrep (Metuchen, NJ, USA). The chemical composition analysis was performed using a MagiX PW2424 spectrometer produced by PANalytical (Malvern Panalytical Ltd., Enigma Business Park, Grovewood Road, Malvern, UK) calibrated using a series of certified reference materials JRRM 121-135, JRRM 201-210, and JRRM 301-310 (The Technical Association of Refractories, Japan (TARJ), Xiamen, China).

#### 2.1.2. Grain Size Distribution

Two devices were used for the research: the LS 13 320 Particle Size Analyzer Beckman Coulter (Life Sciences Division Headquarters, 5350 Lakeview Parkway S Drive, Indianapolis, IN, USA) and mainly a Horiba LA-300 device (HORIBA, Ltd., Miyanohigashi, Kisshoin, Minami-Ku, Kyoto, Japan). The grain analysis was performed in an accredited laboratory on the basis of the ISO 13320: 2009 [66] laser diffraction method. This method is applicable to particle sizes ranging from approximately 0.1 μm to 3 mm. The study was conducted in isopropanol. This instrument can accurately measure particle sizes in a range of 0.1–600 microns. The Mie mathematical model was used to calculate the particle size distribution. The measurement characteristics were as follows: dispersant-isopropanol; ultrasounds, 5 s; Laser T%, 86.0.

#### 2.1.3. Thermal Analysis (DTA/DTG/TA)

The clay specimen was tested using an STA 409 PC thermal analyser by NETZSCH (Netzsch Gerätebau GmbH, Wittelsbacherstr. 42, Selb, Germany) coupled with a QMS 403 C Aëolos quadrupole mass spectrometer (Netzsch Gerätebau GmbH, Wittelsbacherstr. 42, Selb, Germany). Simultaneous differential thermal analysis (DTA), thermogravimetric (TG) analysis, differential thermogravimetric (DTG) analysis, and measurement of gases separated from the sample (EGA) were performed. The sample (33.0 mg) was placed in an Al_2_O_3_ crucible and heated from 40 °C to 1000 °C at a rate of 10 °C/min with an airflow of 30 mL/min.

#### 2.1.4. Strength Test

The strength test was carried out according to PN-EN 196-1 [67], which concerns the strength of cement mortars. The test on bending and compressive strength was conducted in a Controls apparatus (CONTROLS S.p.A., model 65, Cernusco sul Naviglio, Italy), and the results were automatically given after the device was properly set.

#### 2.1.5. Leaching Test

PV glass and mortars were leached to determine their harmfulness to the natural environment after binding in a geopolymer matrix. The water extract was made in accordance with PN-EN 12457-2: 2006 [68] (for ACC) and PN-EN 12457-4: 2006 [69] (for mortars) with the ratio of liquid to solid phase = 10 L/kg. Metals were marked with the ICP-OES method according to PN-EN ISO 11885: 2009 [70]. Chlorides and sulphates were determined according to PN-EN ISO 10304-1:2009 + AC 2012 by ion chromatography [71]. Mercury was tested according to PN-EN ISO 12846: 2012 using the CVAAS method (cold vapour ion absorption spectrometry) [72]. The leaching tests were performed in an accredited external laboratory using approved methods.

#### 2.1.6. Total Carbon and Sulphur Examination

Prior to testing, the sample of fly ash was ground to a grain size below 63 µm using a grinder with a tungsten carbide lining and then dried at 105 °C to a constant weight.

The carbon and sulphur content was determined using a Leco SC 144 DR sulphur and carbon analyser by Leco (3000 Lakeview Ave. St. Joseph, MI, USA) with a resistance furnace. The test sample was burnt at a temperature of 1350 °C in a stream of oxygen. The content of CO_2_ and SO_2_ in the resulting gas was determined by measuring the absorption of infrared radiation.

#### 2.1.7. SEM (Scanning Electron Microscopy) Examination

The morphology of the tested sample was determined using a TESCAN Mira 3 LMU scanning electron microscope equipped with an EDS from Oxford Instruments (TESCAN, Abingdon, UK), supported by Aztec software. The specimens were carbon-coated using a Quorum Q150T ES sputter (Quorum Technologies Ltd., Guelph, ON, Canada). The tests were conducted on fractures or powdered material using the SEM-BSE (backscattered electron detector) operating mode or SEM-SE (secondary electron detector). The activation energy of fluorescent radiation used for SEM-EDS analysis was 15 keV.

#### 2.1.8. XRD: X-ray Diffraction

For phase composition analysis, an X’Pert Pro MPD X-ray diffractometer, produced by PANalytical (Westborough, MA, USA), was used. The measurements were conducted at room temperature using monochromatic Cu Kα radiation. Qualitative analysis with the support of the ICDD PDF4+ database was performed employing HighScore v4.9 software (Malvern Panalytical Ltd., Malvern, UK, 2020). The test was conducted on powdered samples. Quantitatively, the phases were determined by the Rietveld method using Siroquant computational program (40 Hoskins St., Mitchell, ACT, Australia).

### 2.2. Characterisation of Components

Materials for the design of binder mixtures with their chemical composition are presented in Table 1. To determine the chemical composition of used components, the XRF method was involved.

#### 2.2.1. PVG: PV Glass

Glass from photovoltaic panels came from a recycler (Thornmann Recycling Sp z o.o., Toruń, Poland) in the form of crushed pieces up to 1 cm in size. The glass was not contaminated with a black layer of encapsulant and solar cells. Chemically, it can be classified as soda-lime glass powder. The cross-section and structure of the panel are shown in Figure 2. To prepare the geopolymeric mixtures, the glass was ground, sieved, and tested in a particle size analyser (Beckman Coulter). The results of PVG (PV glass) powder have the following parameters: mean, 97.19 μm; median, 85.67 μm; d_10_, 4.669 μm; d_50_, 85.67 μm; d_90_, 200.5 μm; and with phases amounts <3%: 1.284 μm; <25%, 25.08 μm; <50%, 85.67 μm; <75%, 166.0 μm; and <97%, 227.8 μm (Figure 3).

#### 2.2.2. KC: Kaolinite Clay

In this examination, kaolin clay (SEDLECKY KAOLIN a.s. 362 26 BOZICANY, Czech Republic, mining date: 15 April 2021) in a powdered form was used. Based on the DTA/TG test result shown in Figure 4, kaolin clay (designated as KC-0) was burned at 600 °C for 1 h in a laboratory ceramic furnace to prepare metakaolin powder specimen, designated as KC-1 in this article.

The mass spectrometer recorded m/e lines from 1 to 70. The thermogram showed a loss of sample mass by 11.13% related to the separation of water from the sample. The maximum water release was at a temperature of 537 °C. The process reached the highest speed at a temperature of 527 °C (minimum on the DTG curve). The weight loss was accompanied by an endothermal heat effect with a minimum on the DTA curve at 528 °C. The temperature range corresponded to the dehydroxylation reaction. The selected temperature of 600 °C for dehydroxylation of the used clay was chosen to ensure that complete dehydroxylation took place, taking into account the rising part of the graph. At temperatures 339 °C and 464 °C in the mass spectrum of the released gases, there were small maxima of the released CO_2_ from the sample.

During the microscopic observations (comparable with [54]), it was noticed that the KC-1 clay, after the calcination process, had a more developed specific surface due to a slightly larger number of smaller agglomerates and separated plates (Figure 5b) than non-sintered kaolin clay KC-0 (Figure 5a). The background is a carbon tape.

The XRD analysis of the KC-0 sample showed the highest amount of not only kaolinite but also quartz, muscovite, montmorillonite, and dawsonite (comparable with [54]). However, the KC-1 sample is mainly the amorphous phase; quartz and KAl_3_Si_3_ (muscovite) were identified. In the KC-0 sample, the main phase is kaolinite, which means that in the KC-1 sample, the metakaolin is amorphous. There is also muscovite after dehydroxylation in KC-1, which has retained its structure.

#### 2.2.3. ALC: Alumina-Lime Cement

Alumina-lime cement was used because of its high calcium and aluminium content. Calcium is needed for pozzolanic reactions, and aluminium is needed to enrich the mixture with this element. It was produced from two raw materials, bauxite and limestone, by sintering at a temperature of 1250–1550 °C in a rotary kiln. According to the XRD examination, the main phase in this cement is CA and the accompanying phases are C_4_AF, C_12_A_7_, and C_2_AS (where C = CaO, A = Al_2_O_3_, F = Fe_2_O_3_, and S = SiO_2_). Its production is controlled according to the PN-EN 14647 standard. Unfortunately, this component is not an anthropogenic mineral but is needed to increase the amount of aluminium and calcium in the mixtures.

#### 2.2.4. GGBFS: Ground Granulated Blast Furnace Slag

Ground granulated blast furnace slag (GGBFS) by Górażdże Cement S.A. (Chorula, Poland), which came from the Ekocem production plant (Dąbrowa Górnicza, Poland), met the standard requirements according to PN-EN 15167-1 [73]. Its specific surface area according to Blaine was 3850 cm^2^/g, and it was delivered and stored in a sealed container. The use of slag as a binder involves the use of an alkaline activator. Together with sodium metasilicate, it acts as a binder in the mixture [13]. Using X-ray diffraction, it was confirmed that the slag contained approximately 98% amorphous phase.

Figure 6 shows the phase composition of the used GGBFS. The main component of the slag was the amorphous phase and a small amount of calcite. Other typical phases of blast furnace slag were also identified from the group of silicates—melilites, i.e., gehlenite and merwinite (island silicate) [74]. There was also a hydrated basic aluminium magnesium carbonate—hydrotalcite. Hydrotalcites (HTCs) are a class of high-temperature chemical sorbents that have been widely investigated for application in sorption-enhanced reactions [75].

#### 2.2.5. ACC: Autoclaved Cellular Concrete

Autoclaved aerated concrete is made of cement, lime, gypsum, aluminium powder or aluminium paste as a blowing agent, sand, and water. The production process of aerated concrete consists of exposing it to saturated water vapour in steam autoclaves. Most often, saturated steam with a temperature of 180 to 190 °C and a pressure of 1.0 to 1.3 MPa is used. ACC, in this examination, was used as a secondary material, and the results of its XRD analysis are shown in Table 2 and Figure 7. Phase identification and quantitative composition were performed using the X-ray diffraction method according to PN-EN 13925-1:2007 (described in the previous paragraph). Elements of concrete blocks were prepared from the non-contaminated remains from the construction site by grinding in a ball mill and screening through sieves. The fraction that passed through a sieve with a mesh of 65 µm was used for the tests.

Particle and cumulative size distribution studies were also performed, obtaining the following parameters: d_10_, 14.61 μm; d_50_, 66.54 μm; and d_90_, 142.40 μm. A Horiba LA-300 device was involved. The examination chart is shown in Figure 8.

#### 2.2.6. FAB: Fly Ash from Biomass Combustion

Fly ash from biomass combustion (FAB) was used as a secondary material; the results of its XRD analysis are shown in Table 3 and point to many different phases. Some of these may come from the contaminants in the soil collected during harvesting. However, the amorphous phase contains more than half of the total mass. Phase identification and quantitative composition were performed using the X-ray diffraction method (described in an earlier paragraph).

Particle and cumulative size distribution studies were also performed, obtaining the following parameters: d_10_, 3.55 μm; d_50_, 19.99 μm; and d_90_, 69.96 μm. A Horiba LA-300 device was involved. The examination chart is shown in Figure 9.

#### 2.2.7. FAC: Fly Ash from Coal Combustion

The examined ash came from the ammonia removal experimental process and contained very large amounts of ammonia, 11.5% of total carbon, and 0.19% of total sulphur. The parameters of the particle and cumulative size distributions were as follows: d_10_, 16.02 μm; d_50_, 106.74 μm; and d_90_, 307.87 μm (Figure 10). A Horiba LA-300 device was involved.

#### 2.2.8. Standard Sand

CEN standard sand was added to the mixture in the amount of 67.2% of the total mass of the mixture in order to make mortars of beams with dimensions of 4 cm × 4 cm × 16 cm for binder strength tests according to PN-EN 196-1 [67].

#### 2.2.9. Alkaline-Activating Solution

An activating solution was prepared using sodium metasilicate pentahydrate Na_2_SiO_3_·5H_2_O, sodium hydroxide NaOH, and distilled water with a conductivity of 0.06 μS in mass proportion 1:2:10.2. The activator was prepared in such a way that the temperature did not excessively rise, i.e., metasilicate was first dissolved using a magnetic stirrer (dissolution is an endothermic reaction), and then sodium hydroxide was added (dissolution is an exothermic reaction). Two types of an activating agent were selected because NaOH is usually used for fly ash and metasilicate for slag [13].

### 2.3. Selection of Ingredients Proportions, Preparing Mixtures

As it is commonly known, the preparation of the geopolymer blend requires the calculation of the ratios of the oxides or elements present in the precursor (Al, Si) and then the quantity and quality of the activator as well as the amount of water. Based on Table 1, a diagram of the mass ratios of selected oxides that are most often analysed in the synthesis of geopolymers is shown in the case of each ingredient (Figure 11).

As can be seen, the composition of the substrates was mainly aluminosilicates with an emphasis on a higher silicon content than aluminium. Even kaolin clay is richer in silica than alumina. This meant that it was decided to additionally use the substrate alumina-lime cement. Calcium is present in greater quantity only in the slag, and this was another reason why ALC was used to create either the C–A–S–H or C–S–H phase. Most of the components contained trace amounts of sodium. Sodium was found only in glass (PVG).

It was decided to prepare 3 types of mixtures, named M5, M6, and M7, in the same proportions but replacing only 1 waste component—autoclaved cellular concrete (ACC) or fly ash from coal combustion (FAC) with higher amount of ammonia ions or fly ash from biomass combustion (FAB). In addition, each of the mixtures was differentiated, in the type of kaolin clay, to calcined and non-calcined to determine the effect of thermal activation of the kaolin clay.

The mixtures M5, M6, and M7 were prepared on the basis of the computational analyses of the chemical composition of the dry components (without the standard sand). The following proportions were proposed—presented in the form of percentage by mass in Figure 12 and in the molar oxide and atomic elemental forms in Table 4. In Table 5, the mass ratios of the components of the mixtures are given.

Based on a polymeric model of Davidovits [45,64], a poly(sialate) type (–Si–O–Al–O–) structure can be expected in the prepared mixtures 2[Al(OH)_4_]^−^ + [SiO_2_(OH)_2_]^2−^, Si/Al = 1, beside the C–S–H phase.

## 3. Results with Analysis

### 3.1. Composition Analysis

It is commonly known that the influences of the molar ratios of SiO_2_/Al_2_O_3_ and Na_2_O/Al_2_O_3_ affect the microstructure and compressive strength of geopolymer binders. In this article, low initial molar ratios of SiO_2_/Al_2_O_3_ (around 2) structures were considered, which favours the condensation of poly-(sialates), with an atomic ratio of Si/Al = 1. High strength and durability systems, in turn, can be achieved when initial molar ratios of SiO_2_/Al_2_O_3_ are around 3.5–4.5 [64].

Juengsuwattananon et al. [64] examined the reaction products of metakaolin-rice husk ash geopolymer with initial molar ratios of SiO_2_/Al_2_O_3_ (2.0–7.0), Na_2_O/Al_2_O_3_ (0.6–1.6), and Na_2_O/SiO_2_ (0.20–0.72). Moreover, the specimens were cured at 30 °C for 1–90 days before the examination of the phase and microstructure and mechanical properties as well. The highest strength was achieved when the initial molar ratios of SiO_2_/Al_2_O_3_ and Na_2_O/Al_2_O_3_ were 4.0 and 1.0, respectively.

Based on the diagram prepared by Juengsuwattananon et al. [64], the composition data for samples M.5.1, M.6.1, and M.7.1 are placed in the red circle in Figure 13. On this basis, it can be assumed that the designed AA-binders were mainly zeolite + geopolymeric phase + unreacted raw materials. However, there were many other compounds in the composition of mortar components, such as calcium, which also have a great influence on the formation of the microstructure and material properties.

### 3.2. Strength Tests

The strength tests were carried out as follows. After 7 days, the bars were demoulded and cured in a laboratory room (22 ± 2 °C) in air. After 7, 14, and 28 days, the bar from each series was tested for its flexural strength. One of the halves was immediately tested for compressive strength, and the other was placed for 3 days in a specialised ventilated dryer for building materials at 105 °C. After these 3 days, the compressive strength of the second half was tested on the 10th, 17th, and 31st days of hardening.

Bending and compressive strength test results for mixtures with KC-0 kaolin clay and KC-1 calcined kaolin clay mortars are shown in Figure 14, Figure 15, Figure 16 and Figure 17.

It can be clearly seen that the calcination of the clay contributed to the increase in the mechanical properties of the mortars. By analysing the graph shown in Figure 14, autoclaved concrete ACC (mortar M.6.0) caused a rapid increase in strength, but in the first 7 days, the bonding reactions were very slow. In the first 7 days, the specimens were not dried out possibly due to water-absorbing clay and curing covered in moulds. As a result of replacing clay with calcined clay, the M.5.1 samples with biomass combustion ash were able to match the strength of the M.6.1 samples with ACC. The least stable was the development of bending strength in the case of samples M.7 with ash with a high content of ammonia.

The obtained compressive strengths of the test specimens generally were also very low, in the range up to 14 MPa (unburned clay) and 19 MPa (calcined clay). The strongly noticeable difference was between the samples heated for an additional 3 days in 105 °C and non-heated ones made of non-calcined clay (Figure 16). It appears that curing at 105 °C has stronger developing strength effects than calcination of clay. In the case of samples with KC-1 clay (Figure 17), the strength achieved on the 28th day under normal conditions was equal to that obtained on the 17th day after 3 days of drying. However, specimens on the 10th day of curing were stronger than on the 14th day of curing. It is not known, however, how this increased hardening temperature will affect the further development of strength, which is another interesting research direction.

### 3.3. Leaching Test

Leaching was prepared in the case of ground PV glass to check that it did not contain harmful metals after the granulation process and in the case of two mortars: mixture designed M.5.1 and M.5.1.1. The M.5.1.1 mixture is the same as M.5.1 (Table 5), but the PV glass content was eliminated to check the influence of PV glass. The mixtures M.5.1 and M.5.1.1 were cured for 28 days in natural conditions and then dried for 3 days at 105 °C. In the total mass of M.5.1, PV glass was just 1.1%. The effects of removing PV glass from the mortar mixture were as follows: better binding of sulphates, better binding of zinc and chromium, but the release of larger amounts of mercury and arsenic (Table 6).

### 3.4. SEM Examination

The microscopic examination consisted of comparing the microstructure of the M.6 series samples in relation to the strength results. Sample M.6.0 was made with non-calcined clay and sample M.6.1 with calcined clay. Both samples were cared for in natural conditions without being dried at 105 ° C. Their endurance was tested on the 28th day of hydration. As can be seen from the photos in Figure 18, these two microstructures differed. The maps of the content of the elements in the studied area show larger oval forms with a high silicon content; these are grains of sand. The 100-fold magnification does not show that the matrix of the binder M.6.0 (Figure 18a) is mechanically significantly weaker than that of M.6.1 (Figure 18b); it is even more uniform. However, at higher magnifications (Figure 18e,f), there is clearly less coherence of its elements. In addition, the ITZ (interfacial transitional zone) was more porous in the case of sample M.6.0 (Figure 18e), which can be seen, in particular, by the slag grain on the right side of the image. The fact that it is a slag grain is evidenced by the distribution of the content of magnesium and calcium (Figure 18g). Perhaps a brighter matrix of M.6.1 (Figure 18b), compared with M.6.0 (Figure 18a), indicates a lower water content in the structure. Too much water can hinder the bonding processes and cause disintegration.

During observations carried out at a magnification of 10 k×, crystalline or other regular forms were not generally visible in the mass. A small piece of the phase morphologically similar to the honeycomb can be seen only in the very centre of the image of sample M.6.0 (Figure 19); probably it is the C–A–S–H phase.

Without wishing to lower the resolution of the photo shown in Figure 19a, the locations of point analyses are shown above in Figure 19b. Table 7 shows the content of the elements with the comparison in the graphs. Probably, in point 22, the C–A–S–H gel in a form similar to a honeycomb can be seen and, in point 23, a chemically active grain surface of SiO_2_.

Elemental analysis of EDS confirmed that hydration products were mainly calcium aluminosilicate hydrate (C–A–S–H) or calcium–sodium aluminosilicate hydrate C–(N)–A–S–H type gels [76] or even with built-in magnesium near slag grains (point 27 in Figure 20a).

The granules seen in Figure 19a are more dissolved in Figure 20a. Fewer voids, larger clusters, and less oblong granules and forming conglomerates with each other, the flake and fibrous phase make the matrix denser in the case of M.6.1 (Figure 20) than in the case of M.6.0 specimen (Figure 19). Similar reports were also noted by Siddika et al. [17]. Therefore, with the fineness in particle size and thermal preparation of KC-0 to KC-1, the microstructure of the AA-binder becomes denser.

Table 8 shows the content of the elements pointed in Figure 20b with the comparison in the graphs. It seems that in point 25, the C–A–S–H gel can be identified; in 26, a glass; in 27, C–A–(Na,Mg)–S–H; in 28 and 29, C–A–S–H; in 30, C–A–S–H enriched with metals (Fe, Ti); and in 31, C–A–S–H from clay flakes.

### 3.5. XRD Examination—Crystalline Phases

M.6.0 and M.6.1 pastes (prepared without the standard sand) marked M.6.0.z and M.6.1.z, respectively, were tested for crystalline phase composition (Figure 21 and Figure 22). Large amounts of sand in mortars can significantly raise the background, and phases present in minor amount will not be noticed. These samples differently matured, however, for 2 weeks in moulds followed by 2 weeks at 105 °C because they were very moist.

The main phase of the samples M.6.1.z and M.6.0.z is the amorphous phase. Other crystalline phases, detected in smaller amounts, may be derived from the components such as gehlenite (from slag), tobermorite (from ACC), or muscovite (from kaolin clay). The following zeolites were identified in the M.6.1.z sample: sodalite (Na_8_[Cl_2_(AlSiO_4_)_6_], PDF#014-6619), cancrinite ((Na,Ca,◻)_8_(Al_6_Si_6_)O_24_(CO_3_,SO_4_)_2_·2H_2_O, PDF#011-2478), and Linde type A (Na_12_[(AlO_2_)_12_(SiO_2_)_12_]·27 H_2_O, PDF#011-3551); and in the M.6.0.z sample, sodalite and MCM-70 (siliceous framework formula Si_12_O_24_, PDF#074-8993) were identified. Zeolitic phases demonstrate the effective formation of an aluminosilicate 3D structure. Cancrinite and sodalite were also identified by Toniolo et al. in soda-lime waste glass in fly ash-based geopolymers [26]. There is no evidence of the formation of the sodium aluminosilicate hydrate (N–A–S–H) type gel, and only calcium aluminosilicate hydrate (C–A–S–H) type gels, in the hardened binder (Si-bearing katoite, PDF#038-0368). Katoite is, in turn, presumably a hydrated CA (CaO·Al_2_O_3_, PDF#008-8343) phase from Ca–Al–cement. There are some differences between the samples; for example, in M.6.0.z, there was no Si-bearing katoite but brownmillerite—a typical phase in the cement composition, which in this composition does not occur in a reacted form. Tobermorite in M.6.0.z was richer in water and keno-structured, as it is known that the C–S–H gel is the major binding phase in alkali-activated slag. In M.6.1.z, there were no potassium-containing phases of aluminosilicates or silicates, unlike in the case of M.6.0.z. Because of dehydroxylation, muscovite became reactive. However, in M.6.1.z (contrary to M.6.0.z), carbonate phases were detected—hydrotalcite and cancrinite. No crystalline sulphate phase was detected in sample M.6.1.z, but the sample had alunite (aluminium potassium sulphate). The sulphates were leachable from ACC and therefore likely reacted with the clay phases. As can be seen, the thermal pretreatment of the kaolin clay greatly contributes to the nature of the crystalline phases also due to the reactions taking place during the alkaline activation.

Phase composition requires an in-depth study also with other methods such as FTIR (Fourier-transform infrared spectroscopy), NMR (nuclear magnetic resonance) spectroscopy, and IR (infrared), which are common methods in the study of geopolymers [52,53,54,61,62,64].

## 4. Conclusions

As Provis [77] stated in 2013, still “geopolymers”—a commercial name for the alkali activation of kaolinite, limestone, and dolomite [78]—attract scientists and entrepreneurs. There has been more and more research in recent years, as we are increasingly focusing on reducing equivalent CO_2_ emissions [79]. Historically, however, it is very important and should also not be forgotten that the alkali activation technology started with a patent obtained by Kühl in 1908 (US Patent 900,939) [77].

This study investigated the effect of incorporating PV glass powder, kaolin clay, ground granulated blast furnace slag, alumina-lime cement, and, interchangeably, an amount of fly ash from coal combustion or fly ash from biomass combustion or granulated autoclaved cellular concrete together in an alkali-activated matrix. As a replacement for kaolin clay, kaolin clay burned in 600 °C was introduced.

Strength tests carried out between the 7th and 31st days of hardening showed very low values, so it is not recommended to use these mixtures in industrial construction. Further research into the composition and care is needed. However, the use of clay after calcination increased the strength of the mortars. The bending strength values most rapidly increased when using ground AAC as compared with using fly ashes in mixtures with non-calcined clay. The samples needed to be treated at a higher temperature because they retained large amounts of moisture. The molar ratios of Na_2_O/SiO_2_ and Na_2_O/Al_2_O_3_ of M5, M6, and M7 mixtures are in line with the values proposed by Davidovits, while the value of SiO_2_/Al_2_O_3_ in the designed mixtures is much lower due to the greater amount of added aluminates, in the case when greater strength is desired. Therefore, a further research stage will be preparing mixtures with the same composition but without the alumina-lime cement.

Preliminary tests showed also that the presence of PV glass in mortars, prepared as described in this work, contributes to the better binding properties of mercury and arsenic. Unfortunately, zinc, chromium, and sulphates more readily released in the company of PV glass powder.

The microscopic observations confirmed the greater densification of the microstructure of the binder made of calcined clay due to its greater surface development and dehydroxylation. The main phase seemed to be C–A–S–H. The AA-binder of non-calcined clay was granular, and the ITZ was more porous. No crystals were found in fractures.

Diffractometry studies of crystalline phases confirmed the formation of zeolites. The research also confirmed that the binder is mainly the amorphous phase, containing C–A–S–H and, in small amounts, C–S–H gels.

Further research requires a longer curing time with temperature control and checking whether in the proposed systems increasing the Si/Al ratio increases the strength. In addition, other research directions have also been suggested because expanding knowledge about geopolymers is nowadays undoubtedly strongly recommended.

## Figures and Tables

**Figure 1 materials-15-05943-f001:**
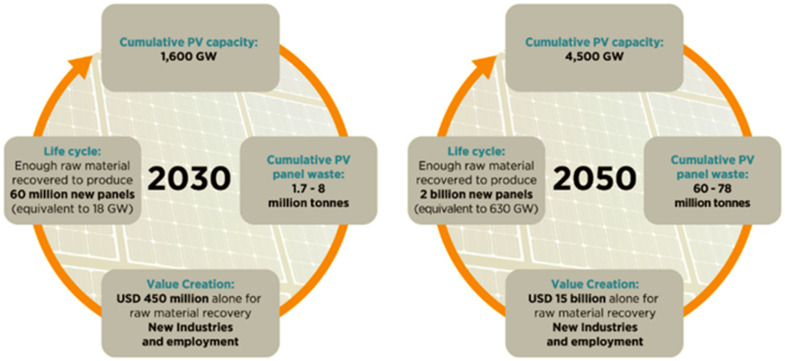
Potential value creation through PV end-of-life management. Source: irena.org [2].

**Figure 2 materials-15-05943-f002:**
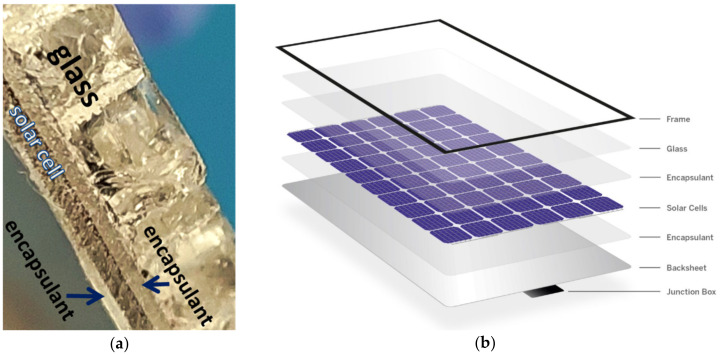
Structure of a photovoltaic panel: (**a**) own photo; (**b**) elements of the construction of a photovoltaic panel. Source: BRIJ http://www.brijencapsulants.com/2020/12/28/six-main-components-solar-panel/ (accessed on 28 December 2020).

**Figure 3 materials-15-05943-f003:**
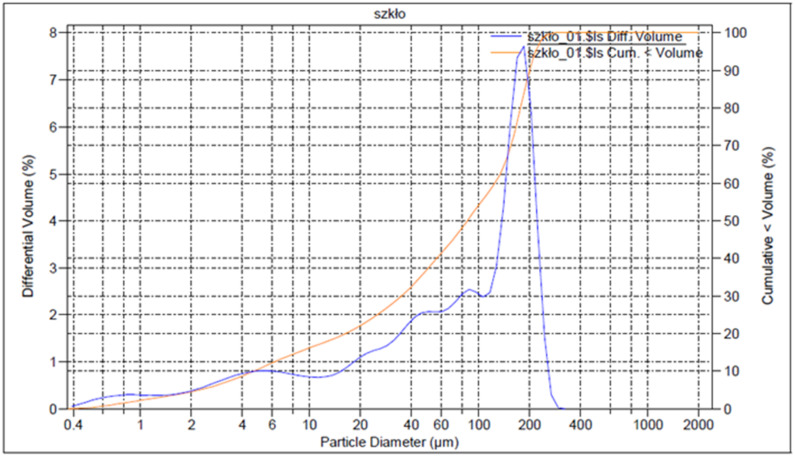
PV Glass powder grain distribution.

**Figure 4 materials-15-05943-f004:**
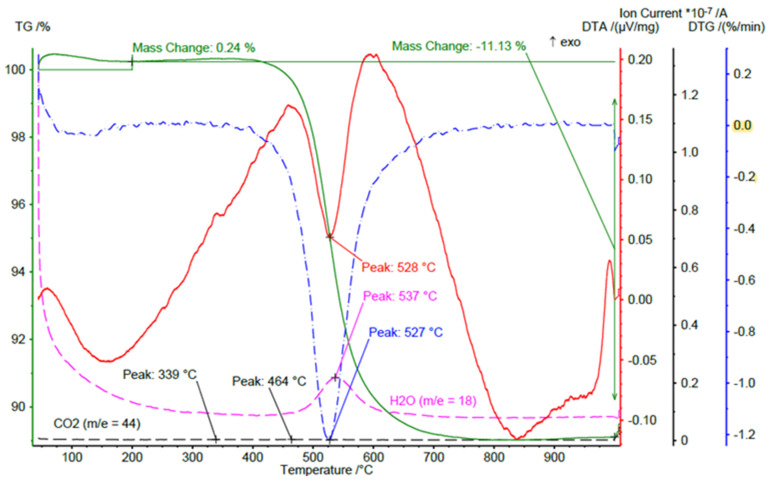
DTA, DTG, and TG analysis of kaolin clay GK-0.

**Figure 5 materials-15-05943-f005:**
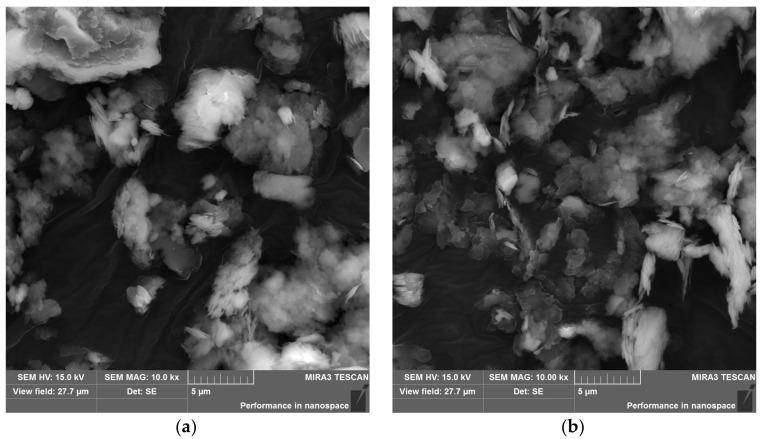
SEM-SE images of (**a**) KC-0 clay and (**b**) KC-1 clay.

**Figure 6 materials-15-05943-f006:**
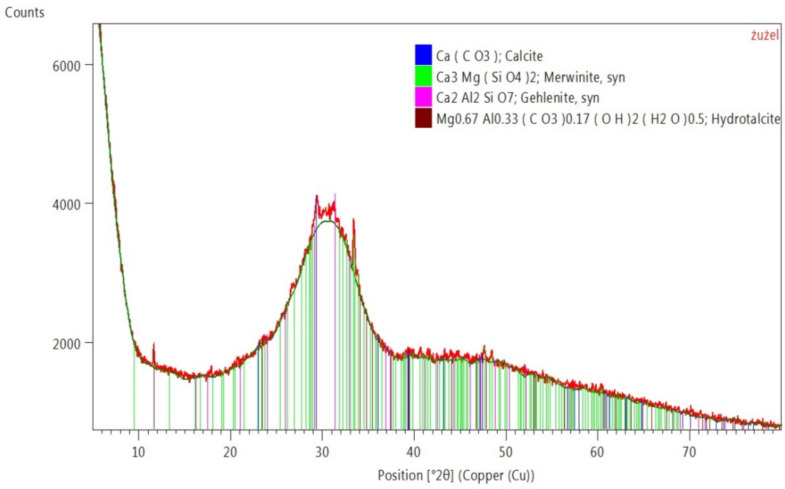
Phase composition of GGBFS.

**Figure 7 materials-15-05943-f007:**
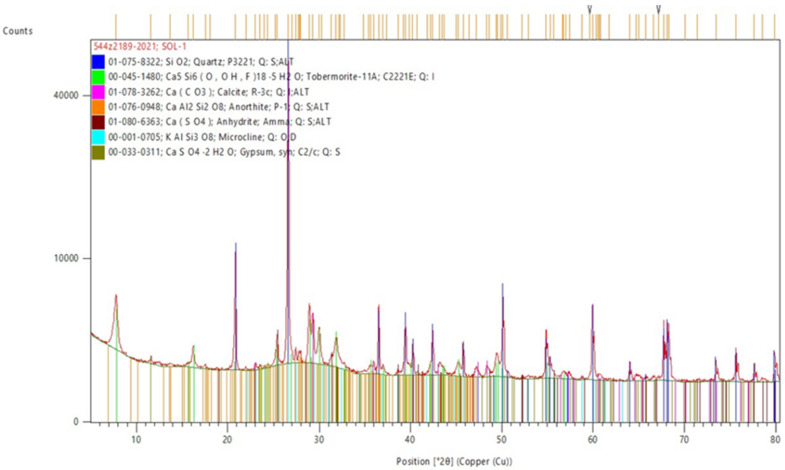
Phase composition of autoclaved cellular concrete.

**Figure 8 materials-15-05943-f008:**
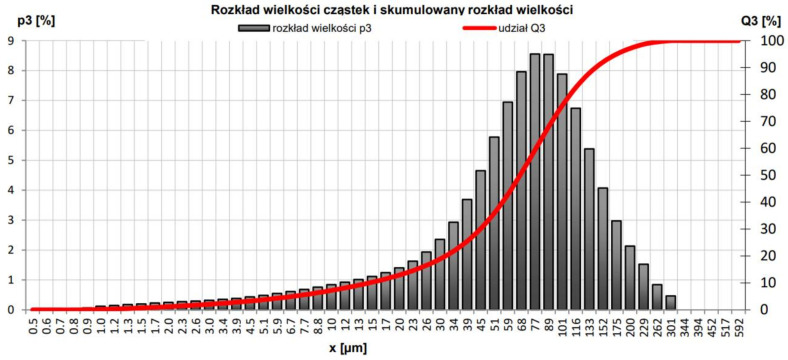
Particle size distribution and cumulative size distribution of ACC.

**Figure 9 materials-15-05943-f009:**
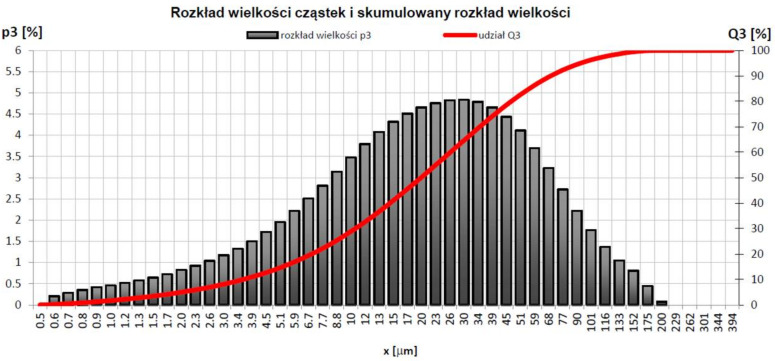
Particle size distribution and cumulative size distribution of FAB—fly ash from biomass combustion.

**Figure 10 materials-15-05943-f010:**
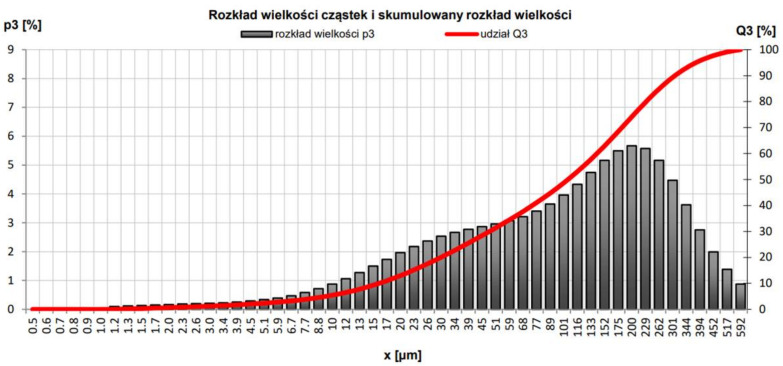
Particle size distribution and cumulative size distribution of FAC—fly ash from coal combustion.

**Figure 11 materials-15-05943-f011:**
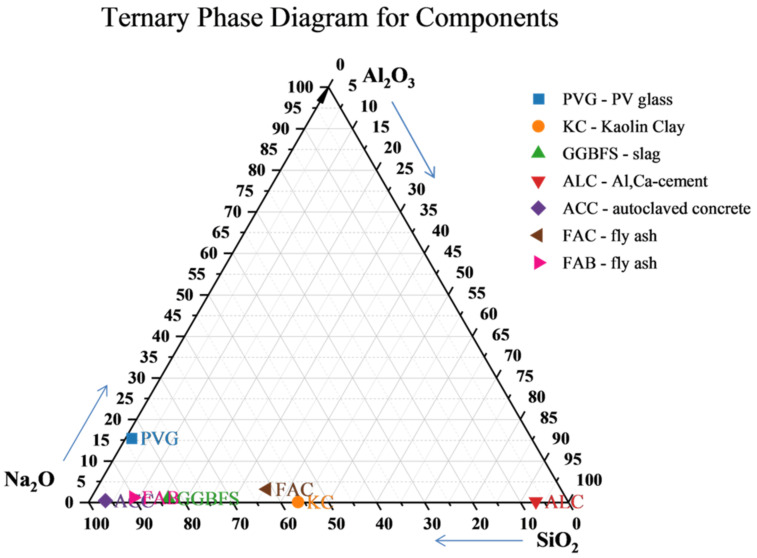
Ternary phase diagram of mass ratios for Na_2_O/Al_2_O_3_/SiO_2_ oxides in components used for mixture preparation.

**Figure 12 materials-15-05943-f012:**
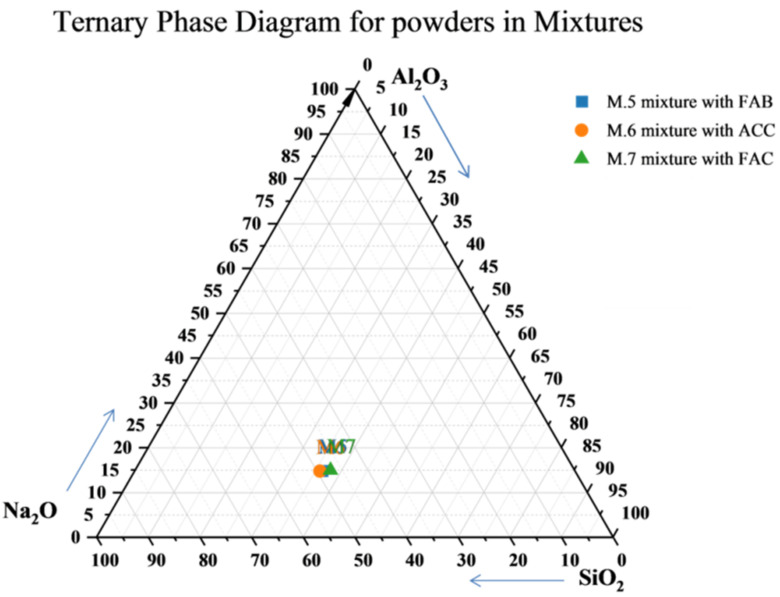
Ternary diagram of mass percentage ratios of Al, Si, and Na oxides in M5, M6, and M7 mixtures including all dry components of binder (without standard sand).

**Figure 13 materials-15-05943-f013:**
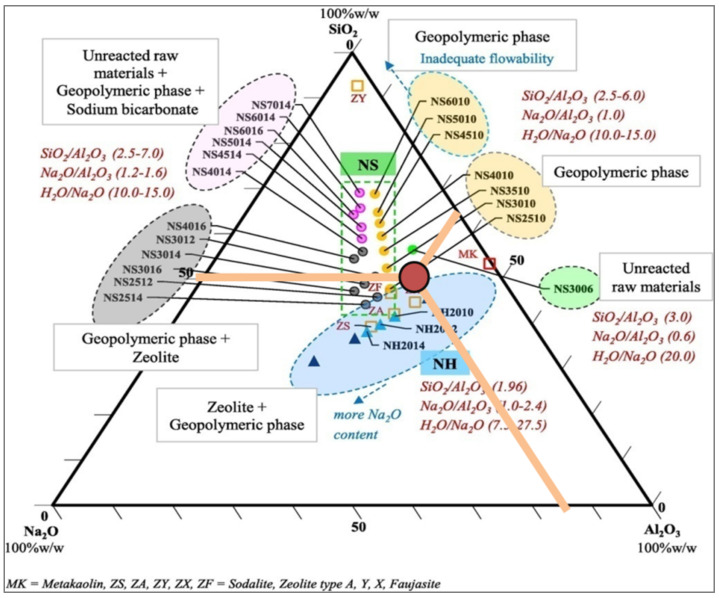
Ternary mixed design composition of geopolymer pastes with different ratios of SiO_2_/Al_2_O_3_, Na_2_O/Al_2_O_3_, and Na_2_O/SiO_2_ from the article prepared by Juengsuwattananon et al. [64] with own mixture data.

**Figure 14 materials-15-05943-f014:**
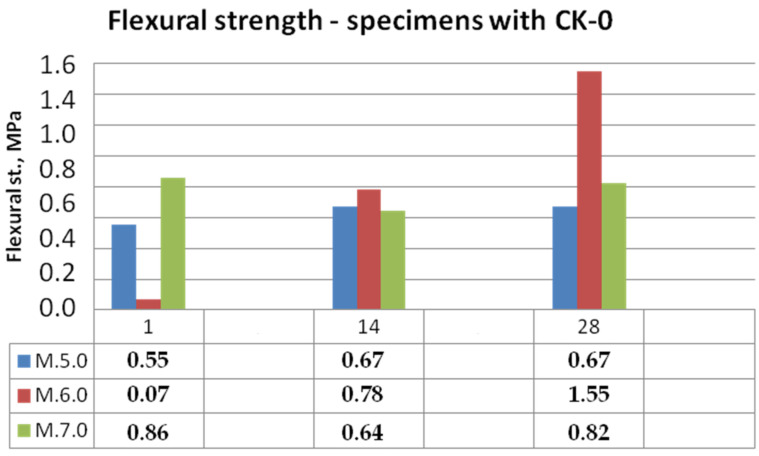
Bending strength of mortars with kaolin clay CK-0.

**Figure 15 materials-15-05943-f015:**
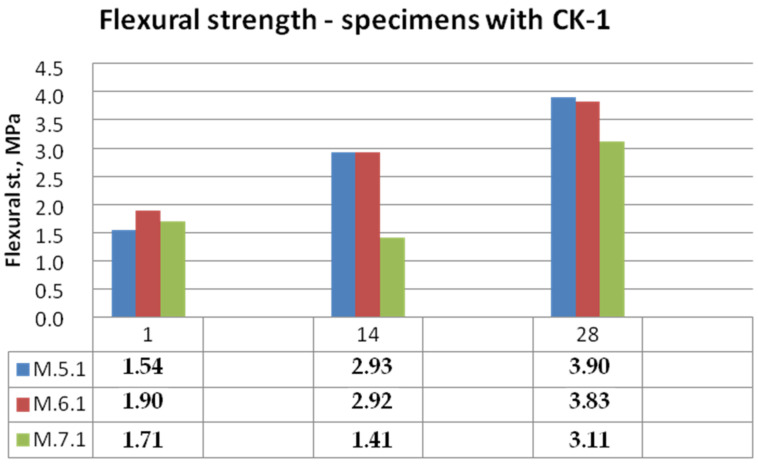
Bending strength of mortars with calcined kaolin clay CK-1.

**Figure 16 materials-15-05943-f016:**
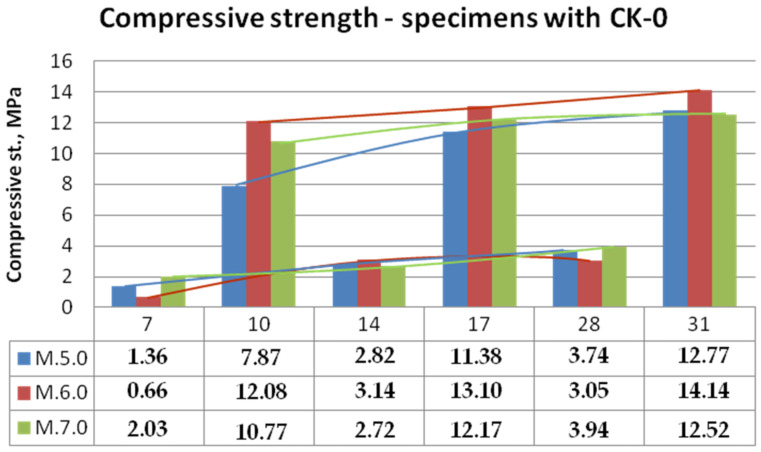
Compressive strength of mortars with kaolin clay.

**Figure 17 materials-15-05943-f017:**
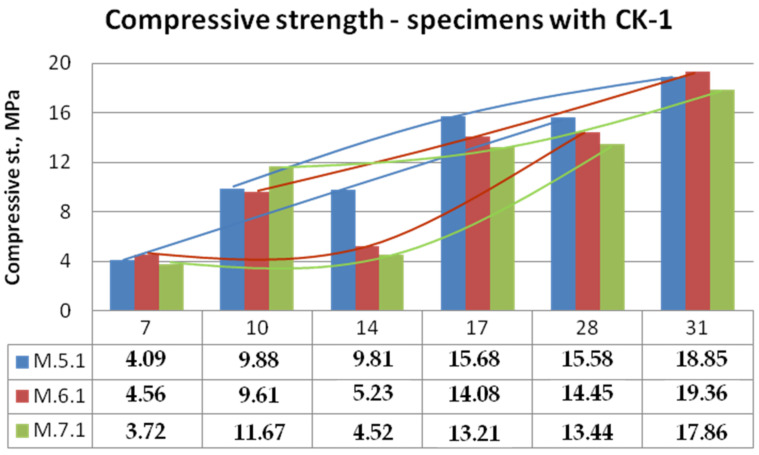
Compressive strength of mortars with calcined kaolin clay.

**Figure 18 materials-15-05943-f018:**
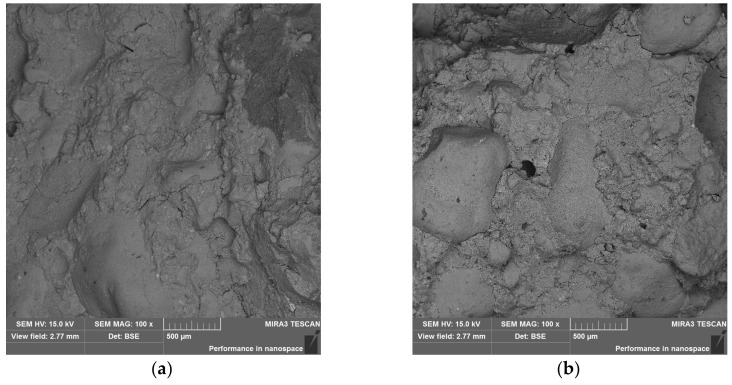
SEM images: (**a**) BSE, M.6.0, mag.: 100×; (**b**) BSE, M.6.1, mag.: 100×; (**c**) map of Si distribution, M.6.0, mag.: 100×; (**d**) map of Si distribution, M.6.1, mag.: 100×; (**e**) BSE, M.6.0, mag.: 1k×; (**f**) BSE, M.6.1, mag.: 1k×; (**g**) maps of elements distribution, M.6.0, mag.: 1k×; (**h**) maps of elements distribution, M.6.1, mag.: 1k×.

**Figure 19 materials-15-05943-f019:**
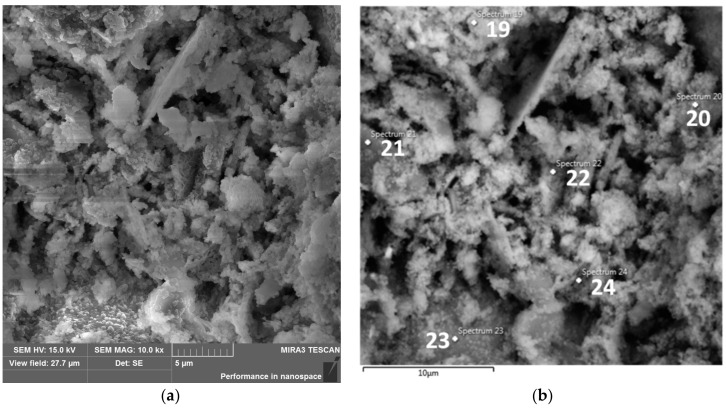
(**a**) SEM-SE image of M.6.0, mag.: 10k×. (**b**) SEM-SE image of M.6.0, mag.: 10k×. Points in which EDS analysis was performed.

**Figure 20 materials-15-05943-f020:**
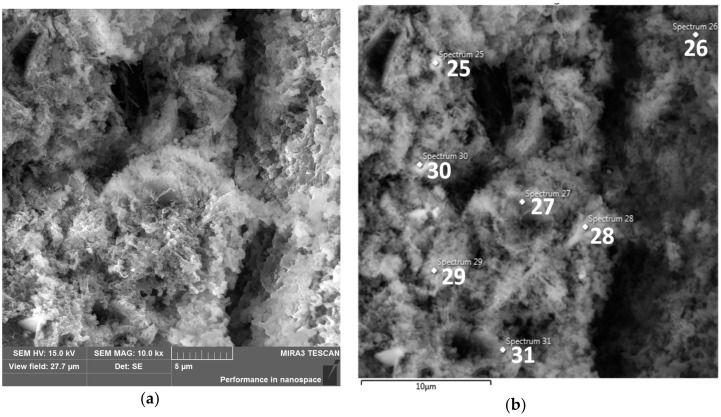
(**a**) SEM-SE image of M.6.1, mag.: 10k×. (**b**) SEM-SE image of M.6.1, mag.: 10k×. Points in which EDS analysis was performed.

**Figure 21 materials-15-05943-f021:**
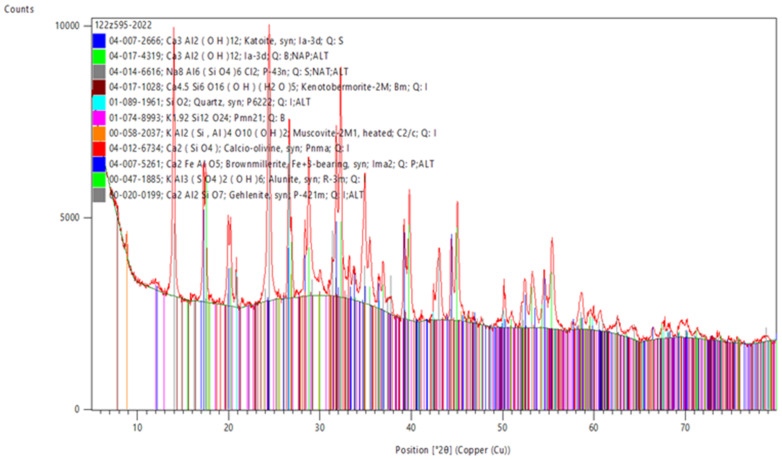
XRD pattern of M.6.0.z specimen.

**Figure 22 materials-15-05943-f022:**
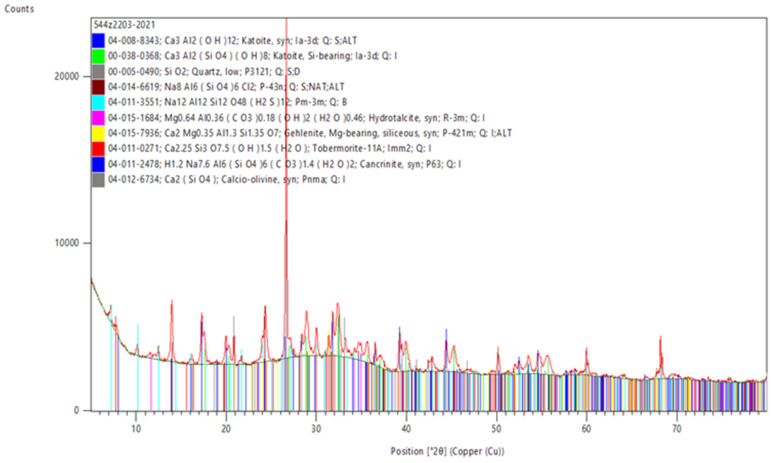
XRD pattern of M.6.1.z specimen.

**Table 1 materials-15-05943-t001:** Chemical composition of components for preparation of binder mixtures.

Component (PL Abbreviation)	PV Glass(S1)	Kaolin Clay	Ground Granulated Blast Furnace Slag	Alumina-Lime Cement (CG-40)	Autoclaved Cellular Concrete (SOL.1)	Fly Ash from Coal Combustion (WR25.1)	Fly Ash from Biomass Combustion (PLB_T1)
Abbreviation used in this article	PVG	KC	GGBFS	ALC	ACC	FAC	FAB
	Content, wt.%						
Silica as SiO_2_	72.39	47.46	40.43	3.05	67.45	43.53	60.25
Aluminium as Al_2_O_3_	1.10	36.75	7.88	42.52	2.32	25.30	5.87
Sodium as Na_2_O	13.41	0.02	0.46	0.08	0.26	2.22	0.72
Calcium as CaO	9.01	0.23	43.27	35.87	17.26	4.42	11.47
Iron as Fe_2_O_3_	0.05	0.92	0.81	15.31	1.00	6.80	2.86
Magnesium as MgO	3.09	0.24	6.97	0.50	0.29	2.61	3.51
Potassium as K_2_O	0.02	0.88	0.29	0.09	0.60	4.67	6.91
Titanium as TiO_2_	0.02	0.24	0.28	2.01	0.06	1.39	0.39
Manganese as MnO	<0.01	0.02	0.16	0.07	0.02	0.07	0.37
Phosphorus as P_2_O_5_	<0.01	0.08	0.02	0.06	0.04	3.41	1.63
Chromium as Cr_2_O_3_	0.01	0.01	0.01	0.08	0.01	-	0.02
Zirconium as ZrO_2_	<0.01	<0.01	<0.01	0.05	<0.01	-	0.04
LOI (550 °C)	0.31						
LOI (1025 °C)		13.35	0.46	0.29	10.44	26.31	5.88

**Table 2 materials-15-05943-t002:** Phase composition of autoclaved cellular concrete (ACC).

Phase	Amount, wt.%
Amorphous phase	26.6
Quartz	38.4
Tobermorite	24.1
Calcite	4.0
Anorthite	1.5
Microcline	2.1
Anhydrite	2.0
Gypsum	1.5

**Table 3 materials-15-05943-t003:** Phase composition of Fly Ash from biomass combustion (FAB).

Phase	Amount, wt.%
Amorphous phase	51.3
Quartz	26.6
Potassium chloride (KCl)	0.5
Potassium chloride (K0.9Na0.1Cl)	0.4
Calcite	2.9
Anhydrite	1.8
Plagioclase (labradorite)	6.6
Feldspare (microcline)	6.7
Mica	0.7
Lime	0.1
Akermanite	1.3
Cordierite	0.3
Leucite	0.6

**Table 4 materials-15-05943-t004:** Molar ratio of Si, Na, and Al oxides and atomic ratio of Si, Na, and Al in the designed mixtures M5, M6, and M7 for binder components.

Mixture	Oxide Molar Ratio			Atomic Ratio		
	SiO_2_/Al_2_O_3_	Na_2_O/SiO_2_	Na_2_O/Al_2_O_3_	Si/Al	Na/Si	Na/Al
Mixture 5	2.28	0.33	0.75	1.14	0.66	0.75
Mixture 6	2.33	0.33	0.76	1.17	0.65	0.76
Mixture 7	2.12	0.34	0.73	1.06	0.69	0.73

**Table 5 materials-15-05943-t005:** Quantitative and qualitative compositions of mixtures, in %.

Component		Type of Mixture with Components Amount. %						% Mass of Binder Component
		M.5.0	M.5.1	M.6.0	M.6.1	M.7.0	M.7.1	%
PVG	PV glass	1.1	1.1	1.1	1.1	1.1	1.1	5
FAB	Fly ash from biomass combustion	1.1	1.1					5
ACC	Autoclaved cellular concrete			1.1	1.1			
FAC	Fly ash from coal combustion					1.1	1.1	
KC-0	Kaolin clay	3.2		3.2		3.2		15
KC-1	Sintered kaolin clay		3.2		3.2		3.2	
GGBFS	Slag	6.5	6.5	6.5	6.5	6.5	6.5	30
ALC	Alumina-lime cement	6.5	6.5	6.5	6.5	6.5	6.5	30
A.1	Na_2_SiO_3_·5H_2_O	1.1	1.1	1.1	1.1	1.1	1.1	5
A.2	NaOH	2.2	2.2	2.2	2.2	2.2	2.2	10
SS	Standard sand	67.2	67.2	67.2	67.2	67.2	67.2	-
DW	Distilled water	11.2	11.2	11.2	11.2	11.2	11.2	-

**Table 6 materials-15-05943-t006:** Water extract of selected components of mixtures, in mg/dm^3^.

Component	PV Glass (PVG)	Specimen M.5.1	Specimen M.5.1.1
Chlorides as Cl^−^	<1.00	<1.00	<1.00
Sulphates as SO_4_^2−^	<1.00	16.6	11.2
Zinc as Zn	<0.001	0.005	0.003
Cadmium as Cd	<0.001	<0.001	<0.001
Copper as Cu	0.005	<0.001	<0.001
Lead as Pb	0.006	<0.001	<0.001
Nickel as Ni	<0.001	<0.001	<0.001
Barium as Ba	0.020	<0.001	<0.001
Chromium as Cr	<0.001	0.029	0.025
Mercury as Hg	0.00020	0.06	0.11
Arsenic as As	<0.01	0.029	0.035
Molybdenum as Mo	<0.02	<0.02	<0.02
Cobalt as Co	<0.01	<0.01	<0.01
Tin as Sn	<0.02	<0.02	<0.02

**Table 7 materials-15-05943-t007:** Content of elements in analysed points, in wt.%.

Spectrum Label	Spectrum 19	Spectrum 20	Spectrum 21	Spectrum 22	Spectrum 23	Spectrum 24
C	2.94	3.84	4.03	1.58	2.89	
O	52.09	55.74	55.36	37.83	56.36	32.86
Na	3.29	4.60	3.60	4.98	0.82	1.62
Mg	0.22	1.01	0.49	2.99		
Al	12.50	13.66	15.00	15.97	1.16	12.27
Si	10.58	12.65	14.99	13.09	37.62	4.84
S	0.12	0.22	0.20	0.44	0.03	0.58
K	0.17	0.12	0.13	0.31		0.13
Ca	14.22	6.88	5.56	18.81	0.78	36.68
Ti	0.53	0.15		0.72		0.38
Fe	3.33	1.13	0.64	3.28	0.33	10.64

**Table 8 materials-15-05943-t008:** Content of elements in analysed points, in wt.%.

Spectrum Label	Spectrum 25	Spectrum 26	Spectrum 27	Spectrum 28	Spectrum 29	Spectrum 30	Spectrum 31
C	3.21	4.87	4.12	3.49	2.72	1.24	5.59
O	51.93	53.71	50.47	54.80	54.12	33.45	51.94
Na	4.65	5.71	7.90	4.01	2.04	4.80	8.66
Mg	0.10	0.07	4.12	0.61	0.42	0.28	2.84
Al	6.48	5.39	11.16	9.18	8.01	6.78	11.96
Si	6.49	27.49	9.81	6.67	5.17	7.53	11.18
S	0.08	0.12	0.23	0.22	0.21	0.18	0.20
K	0.07	0.14	0.14			0.18	0.26
Ca	14.80	2.28	10.06	16.87	20.28	11.53	6.36
Ti	6.12		0.19	0.32	1.62	17.40	
Fe	6.07	0.23	1.83	3.84	5.40	16.63	1.02

## Data Availability

Not applicable.

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
