# Peer review of "Multicomponent Low Initial Molar Ratio of SiO2/Al2O3 Geopolymer Mortars: Pilot Research"

_materials, 2022, doi:10.3390/ma15175943_

Round 1

Reviewer 1 Report

The manuscript entitled " Multicomponent low initial molar ratio of SiO2/Al2O3 geopolymer mortars with PV glass powder, kaolin clay, alumina-lime cement, autoclaved cellular concrete, slag and fly ashes. Pilot research." presents an interesting experimental study conducted on the obtaining and characterization of geopolymers with PV waste addition. However, the paper has multiple issues that must be addressed. The paper needs major revisions before it is processed further, some comments follow:

Title: The title is too long and unclear. Please consider replacing the title with a clear formula that reflects the content of the manuscript.

Abstract: Please remove the unnecessary acronyms from the Abstract. This section must be suitable for separate presentations (independent of the manuscript text body). Therefore, please keep only the acronyms that are used in the abstract and introduce the acronyms for the manuscript the first time when those terms appear in the manuscript text body.

Introduction section

The introduction should be significantly improved. Please conduct a comprehensive and exhaustive study of the previous literature. Please clearly highlight the pros and cons of previous results and justify the need for the current research. Please discuss the highlights individually and assure a clear correspondence between the affirmations from the manuscript and those from the cited papers (the citations introduced in a bulk form (“[4-10]”, [17-30], [23-27] etc.) should be removed and a clear correspondence between the cited studies and the presented information should be provided).

Methods and Materials Section

"The content of CO2 and SO2 in the resulting" – line 222 please move the numbers to subscript. Please make corresponding checking and corrections in the entire manuscript.

Table 1 - two types of iron oxides have been detected in this type of material, therefore, please replace Fe2O3 with FexOy or provide scientific proof to support your results.

2.2.2. KC: kaolinit clay subsection

Please introduce corresponding citations to support your affirmation regarding the thermal analysis.

Figure 5 – please introduce figure labels to highlight the areas of interest for the reader (please indicate those areas observed and described in lines 278-281).

2.2.4. GGBFS

XRD spectra: Why some peaks were considered instead of others. As can be seen in the spectra, some clear peaks around 30 were not evaluated. Please evaluate all the peaks and make corresponding appreciations.

Raw materials- Overall comment.

What was the rationale in making some experiments/determinations for some raw materials and others for other raw materials, i.e., the thermal behavior of metakaolin was analyzed, while for the GGBS the phase composition was studied? Please provide the results for the same analyzing technique to show a clear comparison between these materials...

Results and discussion section

3.2. Strength tests

How many samples have been tested? Please provide the number of samples and the standard deviation values.

The study is more focused on the evaluation of different raw materials and not on its aim of studying the PV effect in geopolymers. Further, the unnecessary information from the materials and methods section should be removed or briefly presented and the authors should better/advanced analyze the interface zone between the PV particles and the geopolymeric matrix. There is the novelty… 

Author Response

Dear Reviewer, I am sending the answers in the attached document.

With gratitude - Barbara Słomka-Słupik.

Reviewer 2 Report

In this study, the low initial molar ratio of SiO2/Al2O3 geopolymer mortars with multicomponent were investigated. The molar ratio of SiO2/Al2O3 played an important role on the quality of geopolymer. And the strength of geopolymer containing sintered kaolin can reach more than 100MPa. The authors used several characterizations such as X-ray diffraction, thermogravimetric analysis and scanning electron microscopy to analyze the products. Although the methods and conclusions in the study were interesting, the following problems still need to be addressed before it can be accepted.

1.     The addition of autoclaved cellular concrete (AAC) and alumina- lime cement (ALC) in this study is confusing. Why?

2.     The aim of this study is the influence of molar ratio of SiO2/Al2O3 on the performance of geopolymer or the content of kaolin clay (sintered kaolin clay) on the performance of geopolymer.

3.     The mineral composition in Fig.6 is missing.

4.     The figures should be clearer. (Such as Fig.5, 6, 7, 8, 9, 10, 11, 12, 21 and 22, Table 7 and 8)

5.     In Fig.14, the M6.0 is better than fly ash. The results need to be further discussed.

6.     The crystalline phases were analyzed by XRD. The amorphous phase should be analyzed by TG and FTIR.

7.     In line 86 and 88, the Al2O3 needs to be modified.

8.     In line 179, the equipment manufacturer needs to be confirmed.

Author Response

(The authors gave the same response as above.)

Round 2

Reviewer 1 Report

The authors adressed most of my comments and the manuscript was improved accordingly. However, some minor corrections must be done before the article is processed further:

" [13,15,19,20,21,26,28,38,40,48,49,50]." There is no need for 12 studies to support this well-known characteristic of geopolymers. Please keep only two or three citations and delete the rest. Also, the other sentences from the paragraph need citations from corresponding studies. Please introduce.

"[24,25,52,53,54]." same comments as above.

Table 1 - two types of iron oxides have been detected in this type of material, therefore, please replace Fe2O3 with FexOy or provide scientific proof to support your results.

Thank you. I cannot find the FexOy in Table 1, but there are 2 forms of LOI because of the temperature of glass softening.

Fly ash contains both Fe3O4 (magnetite) and Fe2O3 (hematite), see https://doi.org/10.1016/j.coal.2021.103746. In the chemical composition, the content of Fe3O4 cannot be determined separately from Fe2O3, therefore, please replace Fe2O3 from table 1 with FexOy or provide the XRD analysis of the fly ash used in the study.

Table 7. Please remove the EDS spectra.

The XRD spectra are of poor quality. Please improve.

Author Response

Good morning.
Dear Reviewer, I hope we have met your requirements. Answers are attached

Yours faithfully, on behalf of the co-authors - B.Słomka-Słupik

Reviewer 2 Report

The questions should be answered carefully.

1.     The addition of autoclaved cellular concrete (AAC) and alumina- lime cement (ALC) in this study is confusing. Why?

2.     The mineral composition in Fig.6 is missing.

3.     The figures should be clearer. (Such as Fig.5, 6, 7, 8, 9, 10, 11, 12, 21 and 22, Table 7 and 8)

4.     In Fig.14, the M6.0 is better than fly ash. The results need to be further discussed.

5.     The crystalline phases were analyzed by XRD. The amorphous phase should be analyzed by TG and FTIR.

Author Response

Good morning.
Dear Reviewer, I hope we have met your requirements. Answers are attached.

Yours faithfully, on behalf of the co-authors - B.Słomka-Słupik

Round 3

Reviewer 2 Report

 I have to reject it  

Author Response

Thank you for your time.

B.Słomka-Słupik